# Hybrid Modeling for Simultaneous Prediction of Flux, Rejection Factor and Concentration in Two-Component Crossflow Ultrafiltration

**Maximilian Krippl [1], Ignasi Bofarull-Manzano [1], Mark Duerkop [1,2] and Astrid Dürauer [1,*]** 

[1] Department of Biotechnology, Institute of Bioprocess Science and Engineering, University of Natural Resources and Life Sciences, 1190 Vienna, Austria; maximilian.krippl@boku.ac.at (M.K.); ignasi.bofarull-manzano@boku.ac.at (I.B.-M.); mark.duerkop@boku.ac.at (M.D.)

[2] Novasign GmbH, 1190 Vienna, Austria

* Correspondence: astrid.duerauer@boku.ac.at; Tel.: +43-1476-5479-095

**Abstract:** Ultrafiltration is a powerful method used in virtually every pharmaceutical bioprocess. Depending on the process stage, the product-to-impurity ratio differs. The impact of impurities on the process depends on various factors. Solely mechanistic models are currently not sufficient to entirely describe these complex interactions. We have established two hybrid models for predicting the flux evolution, the protein rejection factor and two components' concentration during crossflow ultrafiltration. The hybrid models were compared to the standard mechanistic modeling approach based on the stagnant film theory. The hybrid models accurately predicted the flux and concentration over a wide range of process parameters and product-to-impurity ratios based on a minimum set of training experiments. Incorporating both components into the modeling approach was essential to yielding precise results. The stagnant film model exhibited larger errors and no predictions regarding the impurity could be made, since it is based on the main product only. Further, the developed hybrid models exhibit excellent interpolation properties and enable both multi-step ahead flux predictions as well as time-resolved impurity forecasts, which is considered to be a critical quality attribute in many bioprocesses. Therefore, the developed hybrid models present the basis for next generation bioprocessing when implemented as soft sensors for real-time monitoring of processes.

**Keywords:** semi-parametric model; neural network; tangential flow filtration; downstream processing; advanced process monitoring

## 1. Introduction

Membrane separation is a unit operation used in virtually all bioprocesses. One prominent type, crossflow ultrafiltration, is widely used from cell harvest and virus clearance approaches to product concentration steps. In downstream processing of biopharmaceuticals, ultrafiltration (UF) is commonly applied for concentration and buffer exchange after the capture step. It is also applied after virus filtration in single-pass mode to concentrate the sample before it is loaded onto the polishing chromatography, or after polishing to reach the final conditions for product formulation [1]. These process steps entail varying ratios of process and impurities to product concentration.

Modeling of process steps is of increasing importance for bioprocesses. Such process models increase understanding of processes, facilitate the discovery of optimal process conditions and are indispensable for model predictive control. The latter is a cornerstone of Quality by Design and Process Analytical Technology, which is recommended by authorities for biopharmaceutical production. The right balance of model complexity and usability is crucial to employ such models effectively for different unit operations.

To simplify the modeling of downstream processes, a common assumption is to reduce the overall sample composition down to a single target molecule. Coefficients and parameters used in mechanistic models, such as mass transfer models, are often approximated, taking only the target molecule into account. Such models may be limited if the sample contains high levels of impurity.

For some process steps, such as polishing chromatography [2] or ultra/diafiltration [3,4] before formulation, this assumption of one-component solutions is realistic, since the product is already of high purity at this process stage. For earlier process steps, however, this simplification deviates substantially from reality and can lead to erroneous models, e.g., for filtration steps after the capture step. Here, the neglected presence of host cell proteins [5], DNA [6], or protein aggregates [7] can strongly distort the prediction of the model, since effects like membrane fouling and interactions between the product and impurities are not considered. In more complex mechanistic models, if the impurities are well characterized, such effects can be considered. For example, for crossflow filtration, a hard sphere-based mixture model, including multiphase computational fluid dynamics and concentration polarization, was applied to a whey protein solution, leading to a permeate flux prediction error within 20% [8]. Other work has shown that mechanistic models of pore blockage and cake filtration can also predict filter fouling during virus filtration, as a function of the protein of interest, virus and membrane [9]. The initial and late stage of the filtration, however, was dominated by different mechanisms, rendering it difficult to build a valid model for the entire process. The influence of two components on (crossflow) UF was found to affect the process in different ways, from strong [10] to weak [11] to varying [5,12–14] protein-protein (or protein-membrane) interactions. To account for the highly different effects of all components on the process, the experimental part of data generation to estimate the parameters for mechanistic models might become very labor-intensive and the calculations rather complex. Further, if the overall behavior of the process changes because of varying concentrations of impurities, the assumptions of mechanistic models might not hold, to the detriment of the prediction.

One advantage of machine learning supported modeling approaches is that the effects of the impurity on flux and membrane fouling do not need to be fully quantified by the operator [15]. The quantification of these phenomena is performed by machine learning tools, such as an artificial neural network (ANN) [16]. Hybrid models combine the advantages of data-driven black box models (such as ANNs), correlating input with output variables (such as the concentration of impurity with the decrease in flux) with knowledge-based mechanistic models (white box models) derived from conservation of kinetic laws [17]. Hybrid models have been applied to bioprocesses for upstream [18] and downstream applications [19,20].

To compare the predictive power of a model concerning the training space, two terms are often used: interpolation and extrapolation. Interpolation allows the model to make predictions for parameters that lie within the range of training experiments. A model with good interpolation capabilities can make predictions with fewer training observations, since it is able to make reliable estimates of the spaces between the observations. A model with poor interpolation capabilities requires more granular coverage of the training space to make accurate predictions of test experiments. Extrapolation (also called range extrapolation) describes the extent to which a model can make predictions if the tested input parameters are outside the training space. A model with good extrapolation capabilities can make accurate predictions for parameters beyond the training space. A detailed explanation of interpolation and extrapolation in hybrid modeling is given in [21].

Recently, we have shown the benefits of hybrid modeling for the prediction of UF flux evolution. However, this previous model was only established for a one-component system [22]. In the present study, we extended the hybrid model to describe the impact of a modeled protein impurity on the decrease of the permeate flux over time in crossflow UF including the rejection behavior of the product and the impurity. This enables the operator to gain a more detailed understanding of the process. In addition, the impurity concentration is a critical quality attribute (CQA) in almost all manufacturing bioprocesses and if it is too high, the produced batch must be discarded. The presented hybrid models can predict the impurity concentration up front, and potentially minimizes the risk of batch rejection.

Product and impurity were mimicked with different ratios of bovine serum albumin (BSA) to lysozyme concentrations in the starting solution. BSA and lysozyme exhibit different physicochemical properties to facilitate separation and quantification. While BSA was fully retained by the membrane, lysozyme was only partially retained, rendering the predictions of the permeate flux over time even more complex. In a first assessment, we compared the abilities of the well-established mechanistic stagnant film model (SFM) and the recently established one-component hybrid to predict the filtration progress of a two-component solution. Further, we presented two hybrid model structures to predict the evolution of permeate flux and protein concentration of product and impurity by multi-step ahead predictions. One hybrid model included a static lysozyme rejection factor ($R_{Lys}$), while the other updated $R_{Lys}$ dynamically in an iterative way. These model outputs were influenced by the transmembrane pressure (TMP), crossflow velocity (CF), the initial BSA concentration $c_{B,BSA}$ and lysozyme bulk concentration $c_{B,Lys}$. Finally, these novel hybrid model structures were compared to the SFM regarding flux and concentration prediction.

## 2. Materials and Methods

### 2.1. Equipment and Chemicals

All UF experiments were performed on an ÄKTA Crossflow system (Cytiva, Marlborough, MA, USA) controlled by UNICORN 5.31 software. The reservoir tank held up to 1100 mL of bulk solution. The system featured an inline pH probe and UV monitor on the permeate side and a pressure-based reservoir level sensor. The experiments were performed with a Sartocon Slice Hydrosart Cassette hydrophilic, stabilized cellulose-based membrane (Sartorius AG, Göttingen, Germany) with a membrane area of 200 cm$^2$. The model proteins were BSA and lysozyme (A2153 and L6876, both purchased from Sigma-Aldrich, St. Louis, MO, USA). The molecular weight cutoff (MWCO) of the membranes was 30 kDa, chosen so that BSA (66 kDa) was fully retained and lysozyme (14 kDa) was partially retained. BSA and lysozyme were chosen to mimic the protein of interest and process-related impurities, respectively. A filtration buffer of 50 mM phosphate-buffered saline (PBS), pH 8, was used.

### 2.2. Training and Test Data Generation

For the training experiments, the bulk reservoir was filled with 1000 mL of the lowest bulk BSA and/or lysozyme concentration $c_{B,BSA}$ and $c_{B,Lys}$ (see Table A1). The following two steps were then alternated. First, the TMP and CF were increased stepwise, while the permeate was redirected to the feed reservoir to keep the protein concentration $c_B$ constant. For each combination of TMP and CF, the permeate flux was recorded. Second, the sample was concentrated until the next desired $c_B$ was reached. These two steps were repeated at all concentrations given in Table A1. A total of 90 equilibrium fluxes were recorded for different concentrations and combinations of TMP and CF. Our previous work with a one-component system [22] showed that this training set size was sufficient to develop a well-trained hybrid model with accurate flux predictions. A detailed summary of all scouted TMPs, CFs, $c_B$s and recorded fluxes is given in Table A1. Samples were taken after each concentration step for offline measurement. A more detailed description of the methodology for the training experiment is given in an earlier publication [22].

During the test experiments, samples were taken from the retentate and permeate. The measured retentate and permeate concentrations were used to calculate the rejection factor R of the model proteins. A summary of the performed test sets is provided in Table A2.

### 2.3. Concentration Polarization Correction

When concentrating the sample throughout the training experiments, we observed that the measured $c_{B,BSA}$ was lower than the expected concentration calculated from permeate volume ($V_P$) and mass balance. The difference between observed and calculated concentration increased with concentration (see Figure A3B). This was because the concentration polarization (CP) layer—the

protein gradient that forms on the surface of the membrane—increased with $c_{B,BSA}$. This deviation was considered for the test experiments by employing a quadratic polynomial function (Equation A1) and used to correct the calculated $c_{B,BSA}$.

*2.4. Protein Quantification*

BSA and lysozyme concentrations were determined with an analytical high-performance size-exclusion chromatography (SEC-HPLC) using a TSKgel G3000SWXL column (5 μm, 7.8 × 300 mm; TOSOH, Shiba, Tokyo, Japan). The separation was performed under isocratic conditions with 50 mM sodium phosphate, 200 mM NaCl, pH 6.5 as running buffer at a flow rate of 0.4 mL/min. Samples were diluted to a final concentration of 0.1 to 1.0 g/L using 50 mM PBS, pH 8 and filtered through a 0.22 μm Millex-GV Filter (Merck Millipore, Billerica, MA, USA) prior to analysis. The injection volume was 10 μL per sample. Due to the difference in the size of BSA and lysozyme, the peaks were fully separated and could be quantified independently, using standard calibrations from BSA and lysozyme stock solutions.

*2.5. Hybrid Modeling*

2.5.1. Black Box Model

The black box inside the first hybrid model (HM 1) aimed to predict the flux based on the combination of inputs parameters: TMP, CF and the bulk protein concentrations of BSA and lysozyme, $c_{B,BSA}$ and $c_{B,Lys}$, respectively. In the second hybrid model (HM 2), an additional black box was employed to predict the rejection factor of lysozyme $R_{Lys}$ (Figure 1B). An ANN was utilized for this purpose and optimized by varying the number of hidden nodes from 1 to 7. The ANN was set up with the feedforwardnet function and trained with the trainbr function, using MATLAB 2018b. A detailed description is the ANN structure and optimizer function is given in the Appendix A.

2.5.2. White Box Model

The white box model is the mechanistic part of the hybrid model and consisted of a mass balance. The incrementally decreasing bulk volume ($dV_B$ in Equation (1)) was derived from the permeate flux (J), which is the output of the black box, and the membrane area (A). The rejection factor R for component i was calculated with Equation (2), considering the concentration of i in both the retentate ($c_R$; in crossflow filtration $c_R$ is equal to $c_B$) and the permeate ($c_P$). Equation (1) and Equation (2) were used to predict $c_B$ of each component and Equation (3) to calculate the $V_B$ after dt.

$$\frac{dV_B}{dt} = -A \cdot J \tag{1}$$

$$R_i = 1 - \frac{c_{P,i}}{c_{R,i}} \tag{2}$$

$$\frac{d(c_{B,i} \cdot V_B)}{dt} = (A \cdot J \cdot c_{B,i})^{R_i} \tag{3}$$

2.5.3. Training and Test Data

$R_{Lys}$ was calculated from the training set with the UV absorbance at 280 nm on the permeate side. A separate lysozyme training run was performed to correlate the UV signal at 280 nm with the permeate concentration determined by SEC-HPLC. The correlation curve (Figure A3A) with an $R^2$ of 0.97 was used to calculate $c_{P,Lys}$, and subsequently $R_{Lys}$ for all observations of the training set was used to train the black box.

The observed flux and $R_{Lys}$ were compared to the predictions of the hybrid models using the normalized root-mean-square error (NRMSE)

$$\text{NRMSE} = 100 \cdot \frac{\sqrt{\frac{1}{n} \sum_{i=1}^{n} \left( y_i - \hat{y}_i \right)^2}}{y_{max} - y_{min}} \qquad (4)$$

where $n$ is the number of overserved fluxes $y_i$ and the corresponding predicted fluxes $\hat{y}_i$. The normalization $y_{max} - y_{min}$ allows a fair comparison of various fluxes due to different concentrations and process parameters.

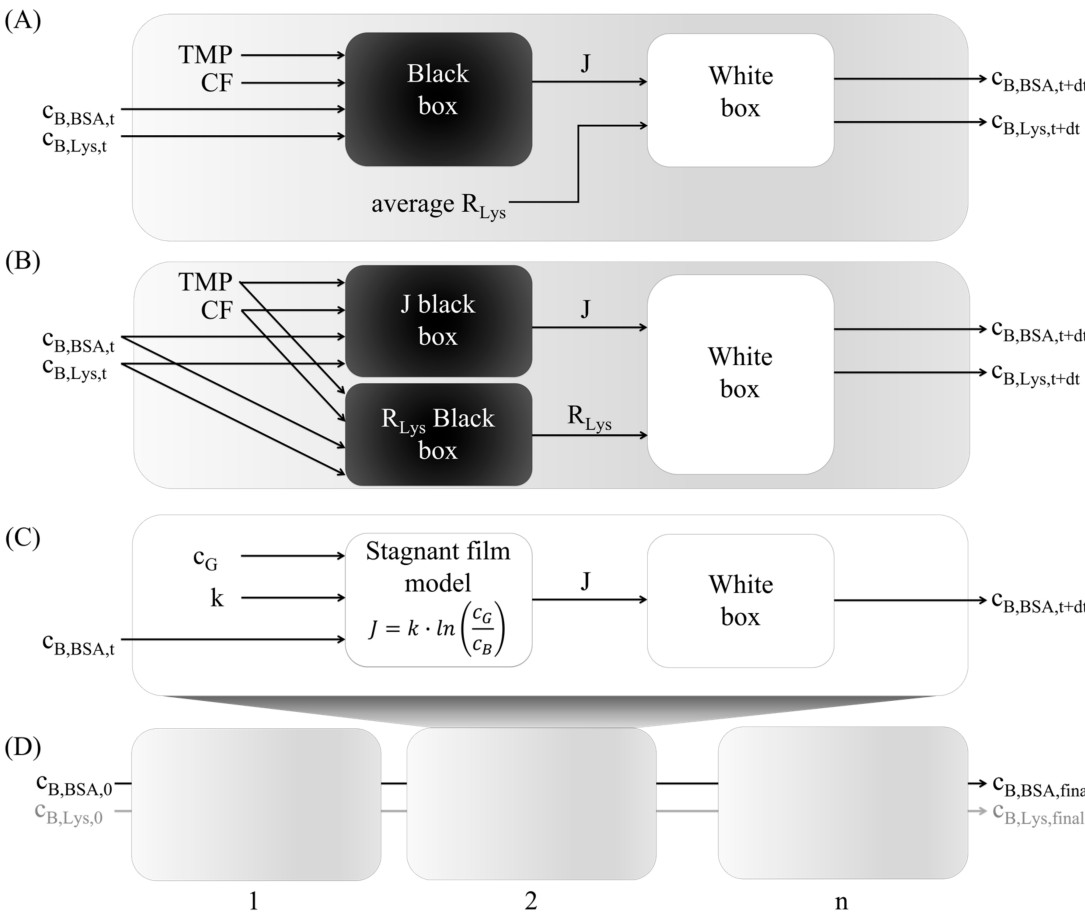

**Figure 1.** Schematic representation of the two hybrid model and mechanistic model structures, with implementation in the multi-step ahead model. (**A**) Hybrid model 1 (HM 1) using static, average $R_{Lys}$ from the training set, (**B**) hybrid model 2 (HM 2) with two separate black boxes for flux and dynamic $R_{Lys}$ prediction, (**C**) stagnant film model (SFM). (**D**) Multi-step ahead hybrid model structure.

2.5.4. Multistep-Ahead Hybrid Model

The structures of the investigated hybrid model are given in Figure 1. The first and simplest HM 1 (Figure 1A) assumed a constant $R_{Lys}$ of 0.77 for all test sets based on the weighted average of all permeate and retentate concentrations samples taken throughout the training experiment. The weighted average considered the variation in $c_{B,Lys}$ and sample intervals using trapezoid rule integration. For the second HM 2 structure (Figure 1B), the flux and $R_{Lys}$ were predicted separately, using two different black box models. The flux and $R_{Lys}$ were fed into the same white box model, which yielded the predicted $c_{B,BSA}$ and $c_{B,Lys}$ after a defined time increment. The developed hybrid model is capable of predicting multiple steps ahead, as depicted in Figure 1D. The multi-step ahead structure uses HM 1, HM 2 or the SFM to predict $c_{B,BSA}$ and $c_{B,Lys}$ for a time increment (dt). The concentrations of the first iteration

were used to calculate future fluxes and $c_B$s of the second iteration, and so on. Multiple iterations were performed until the desired stop criterion was reached. In our case, the stop criterion was the final retentate volume.

The presented hybrid models were used to predict the evolution of flux and $R_{Lys}$ throughout the UF process. Furthermore, the models yielded a prediction for the final $c_{B,BSA}$ and $c_{B,Lys}$. The final $c_{B,BSA}$ and $c_{B,Lys}$ predictions were compared to the final $c_{B,BSA}$ and $c_{B,Lys}$ measured by SEC-HPLC. The model errors were compared using the NRMSE.

Figure 2 shows a flowchart of the hybrid model methodology applied for crossflow filtration. Training experiments were performed by variations in the parameters that are expected to influence the flux. Following this, the model was trained on this training set with a defined experimental design space. The established models were applied to a validation data set that was not used for training. The model structure was optimized by varying the tuning parameters, e.g., number of nodes in an ANN and adding or removing training parameters. The model with the tuning parameters that led to the lowest error in the validation set was then applied to independent test runs with static process conditions.

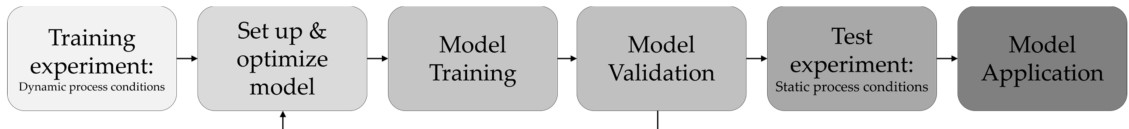

**Figure 2.** Flowchart of the hybrid model methodology for application in crossflow filtration.

2.5.5. Stagnant Film Theory

The presented hybrid models were compared to the established SFM. The SFM derives predictions from the mass transfer model described by convective transport toward the membrane and back-diffusion caused by the concentration gradient [23]. According to the SFM, the flux J is related to the bulk concentration $c_B$ of a single component by

$$J = k \cdot \ln\left(\frac{c_G}{c_B}\right) \tag{5}$$

where $c_G$ is the gel layer concentration at the membrane surface and k is the mass transfer coefficient that depends on the diffusion coefficient and the thickness of the gel layer [23]. The SFM is valid in the pressure-independent region of the filtration. Since k and $c_G$ cannot be adjusted directly during the filtration, a correlation between the adjustable parameters TMP and CF, and k and $c_G$ was required. When plotting the flux versus $\log(c_B)$ for a constant TMP and CF, k and $c_G$ are estimated by the slope of linear regression and $c_G$ was estimated by extrapolating the regression line to the intersection with the abscissa (Figure A6). It has been shown that this way of calculating k yields more accurate results than the Sherwood correlation [24–26] and more solid predictions compared to the osmotic pressure model [27] for similar settings. To compare the SFM to the hybrid models, the black box was replaced by the SFM in Equation (5) using the parameters k and $c_G$ instead of TMP and CF (Figure 1C). In test runs, where the TMP and CF conditions were not covered in the training set, k and $c_G$ were estimated using linear interpolation.

## 3. Results and Discussion

### 3.1. Training Data Description

The data sets for training the hybrid models were generated from filtering BSA and lysozyme with a 30 kDa MWCO cellulose-based membrane (Hydrosart). A total of three training sets were generated covering three CFs (100, 200 and 300 mL/min) and five TMPs (0.8, 1.3, 1.8, 2.3 and 2.8 bar). The three training sets containing BSA, lysozyme and a combination of both are shown in Figure 3. In the

combined training set, the protein concentration of BSA $c_{B, BSA}$ ranged from 3.77 g/L to 77.93 g/L and of lysozyme $c_{B, Lys}$ from 0.28 g/L to 3.81 g/L. The concentration ranges for all training sets are summarized in Table A1. For a better comparison of Figure 3A–D, the *x*-axis of Figure 3C,D are reduced. The entire graphs are given in Figure A2.

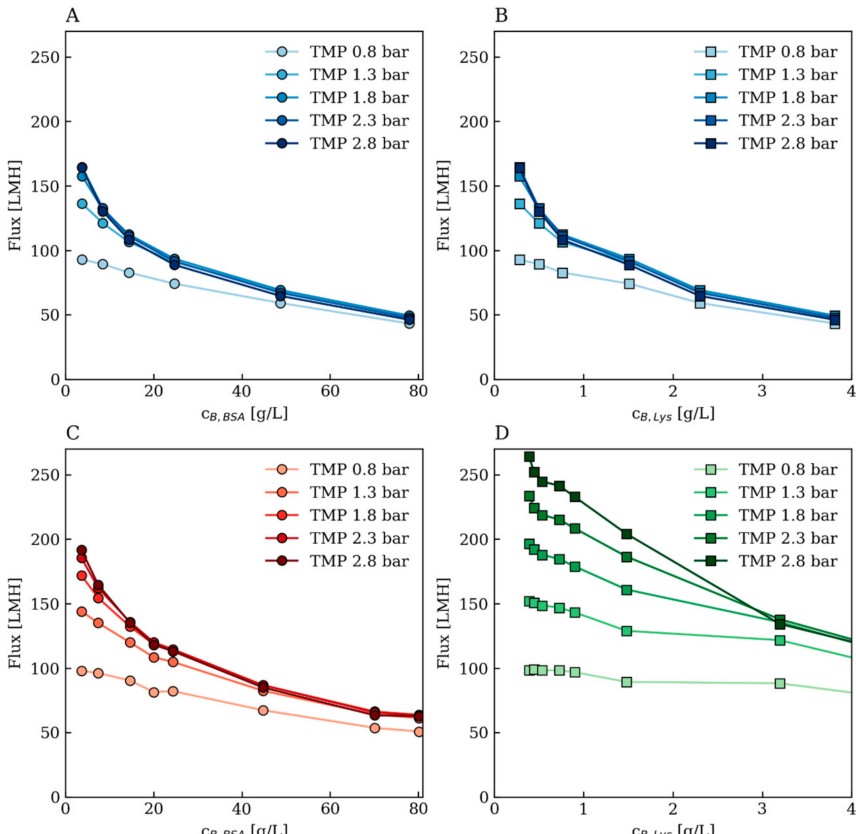

**Figure 3.** Training data sets including different protein concentrations and different TMPs at CF 200 mL/min: two-component training set containing (**A**) BSA and (**B**) lysozyme in the same solution (blue); one-component solution of (**C**) BSA (red) and (**D**) lysozyme (green).

Generally, increasing bulk concentrations $c_B$ led to lower fluxes, while increasing TMP and CF led to higher fluxes in all training sets. This is in accordance with the underlying mechanisms: higher bulk concentrations lead to higher concentrations in the boundary layer and a more prominent effect of the back diffusion along the concentration gradient. An increased TMP leads to higher convective flow towards the membrane, but also to a faster accumulation of the protein at the boundary layer. High CF decreases the thickness of the concentration polarization layer by rectangular displacement. The training set obtained from experiments using only BSA exhibited higher fluxes then the two-component training set. Additionally, the flux decreased faster during filtration of the two-component solution (Figure 3B) compared to the filtration of lysozyme only (Figure 3D). This indicated an increased membrane resistance through the fouling effect on the Hydrosart membrane caused by lysozyme. Being smaller than the pores, lysozyme adsorbed at the inner pore channels [28,29] and reduced its diameter and subsequently the flux through the membrane and the membrane's selectivity.

The two-component training set (Figure 3A,B) was used to train the black box of the hybrid models and to obtain the mechanistic model parameters k and $c_G$. The data set with lysozyme solely (Figure 3D) was used for two reasons: first, to investigate the effect of TMP and CF on the permeability of lysozyme and whether $R_{Lys}$ had to be recalculated for varying input parameters (Figure A5); second, to correlate the permeate lysozyme concentration with the UV signal on the permeate side. This correlation was used to calculate $R_{Lys}$ (Equation (2)) for each observation of the combined training set (Figure 3A,B),

using solely the permeate UV signal. Another training experiment was performed with BSA solely (Figure 3C). The observed fluxes and estimated SFM parameters k and $c_G$ were used to investigate model behavior and error when lysozyme was present in the test set but absent in the training set.

### 3.2. Comparison of the Hybrid Models to the Stagnant Film Theory

The optimal ANN structure in the hybrid models was determined by varying the number of hidden nodes from one to seven and recording the average error of 20 repetitions on the training set. The ANN with four hidden nodes yielded the lowest NRMSE for both HM 1 and HM 2, with an average of 3.4% NRMSE. Higher numbers of hidden nodes led to an error increase due to training set overfitting (Figure A1).

With the SFM, the flux can only be modeled for a one-component system; no adaptations for a two- or multi-component system have been published in the literature so far. In the following, BSA was assumed to be the only component since its concentration was four to 46 times higher than lysozyme in the test runs (Table A2). The k and $c_G$ values of BSA, however, change in the presence of lysozyme. To allow a fair comparison between the hybrid models (which can incorporate multiple components as inputs) and the SFM, both sets of k and $c_G$ were evaluated. Both experiments were carried out with BSA alone. The combination of BSA with lysozyme was used for flux prediction and the results were compared to the prediction of the hybrid models.

The hybrid model trained solely on BSA (Figure 4, red dotted line) and the SFM using k and $c_G$ based solely on BSA (Figure 4, dark grey dot-dashed line) were able to predict a UF process with only BSA present (Figure 4A, black line), but failed to predict the UF flux of BSA and lysozyme (Figure 4B, black line). The latter failed due to membrane fouling by lysozyme and therefore the reduced flux and prolonged process times could not be described by any of these models.

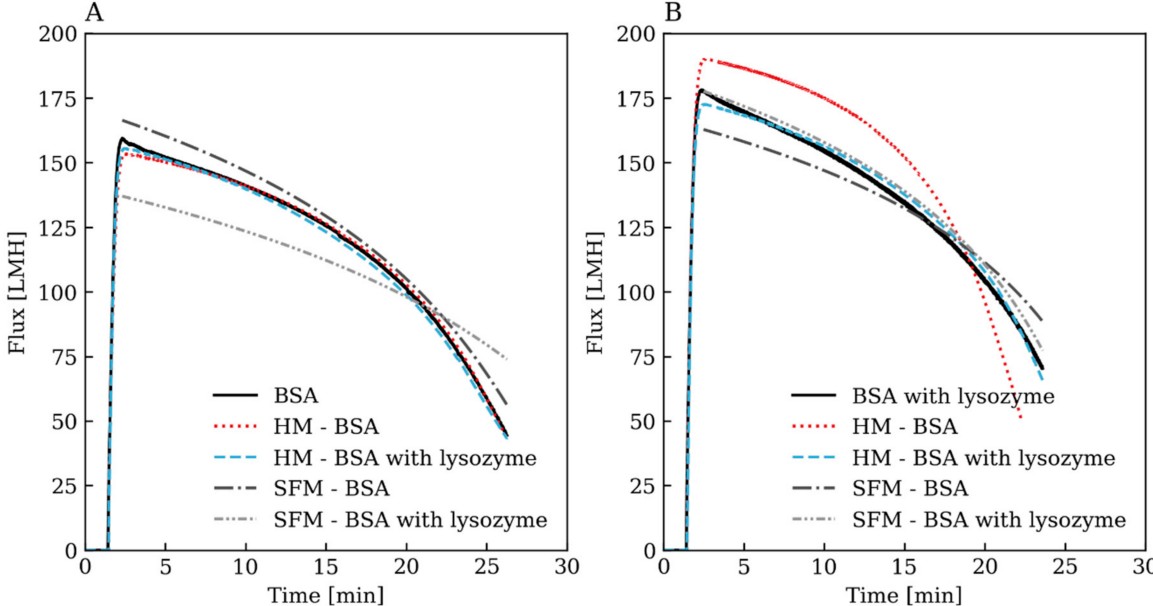

**Figure 4.** Comparing flux prediction of the test set containing (**A**) BSA (TMP 1.8 bar, CF 200 mL/min, initial $c_{B,BSA}$ 6.68 g/L) and (**B**) BSA with lysozyme (TMP 2.1 bar, CF 250 mL/min, initial $c_{B,BSA}$ 3.71 g/L, initial $c_{B,Lys}$ 0.38 g/L), with: hybrid model HM 1 trained on BSA solely (red dotted line) and the BSA and two-component training set (blue dashed line); SFM based on BSA solely (dark grey dot-dashed line) and two-component training set (light grey dot-dot-dashed line).

In contrast, the hybrid model trained with BSA (Figure 3C) and BSA with lysozyme (Figure 3A,B) training runs (Figure 4, blue dashed line) were able to predict both UF processes: BSA solely and BSA with lysozyme (Figure 4, black lines). These results showed that already low amounts of lysozyme

drastically changed the initial flux and flux evolution of the UF process and that incorporating both components in the model was essential for accurate predictions. On the contrary, SFM based on the training run containing BSA with lysozyme was also able to predict the two-component test run well (Figure 4B, light grey dot-dot-dashed line), but showed a drastic offset when predicting a test run with only BSA (Figure 4A, light grey dot-dot-dashed line). The k values from the two-component training set (Table A5) were generally lower than those calculated from solely BSA, since membrane fouling due to lysozyme was assumed. In the absence of lysozyme, however, no membrane fouling occurred and the flux for the same $c_{B,BSA}$ was higher.

In summary, the HM could predict both scenarios, since the varying concentration of lysozyme and its influence on the membrane fouling was integrated into the black box. However, the SFM only predicted one scenario well, depending on which k and $c_G$ were used. For the following two-component predictions, the SFM parameters were based on the two-component training set.

### 3.3. Comparison of Hybrid Model Performance

To further investigate both the interpolation and extrapolation capability of both HMs and the SFM model, a series of test runs were conducted under conditions that were partially not covered by the training sets. To test the hybrid models based on the two-component training set, additional test runs on BSA solutions spiked with lysozyme were performed. The two established hybrid model structures were compared for their $R_{Lys}$, flux and final $c_B$ predictions individually. $R_{Lys}$ effects the in-process $c_{B,Lys}$ prediction and subsequently the flux and final $c_{B,Lys}$. Additionally, the two hybrid models were compared to the SFM in terms of flux and $c_{B,BSA}$ prediction. $c_{B,Lys,}$ and $R_{Lys}$ could not be compared, since SFM can be applied to one-component only.

The test data consisted of nine UF runs performed at different TMP, CF, initial $c_{B,BSA}$ and $c_{B,Lys}$ conditions. Test runs 1–4 were performed within the training space. This meant that TMP and CF were within the training parameters (Figure 5A, blue area) and the initial $c_{B,BSA,}$ and $c_{B,Lys}$ was higher than the initial training concentrations (Figure 5B, blue area). The test runs 1, 2 and 9 were performed in the center of the TMP and CF training space (Figure 5A), with test run 9 containing no lysozyme. Test run 3 was performed at the outer limit of the TMP and CF training space, to investigate how the predictions of the hybrid models changed at the border. Test run 4 was performed under TMP and CF conditions not covered by the training set but within the training space, to investigate the interpolation capabilities of the model. Test runs 5, 6, 7 and 8 were performed under conditions that were partially outside the training space, such as initial $c_{B,Lys}$ (8), initial $c_{B,Lys}$ (5, 6) and CF (7), to test the extrapolation capabilities. The test run parameters are summarized in Figure 5 and Table A2.

#### 3.3.1. Flux Prediction

Regarding the prediction of the flux evolution, the two hybrid models performed similarly (Figures 6A,C,E, 7A,C,E and A4A,C,E). Most test run predictions exhibited a small initial offset. At the beginning of the test experiments, the membrane was clean, while during the training set the membrane exhibited some lysozyme fouling and equilibrium of the concentration polarization layer due to the long training process time. This led to an initially underestimated flux. The offset became more pronounced when initial $c_{B,Lys}$ was higher than 0.3 g/L (test runs 2, 3, 4, 5 and 8; Figures 6C,E, 7A and A4A,C), indicating a stronger membrane fouling at this concentration. Even though all hybrid models were trained with $c_{B,Lys}$ higher than 0.3 g/L, the timely increasing membrane resistance due to fouling reached an equilibrium only after several minutes. After this point, the flux was predicted correctly. The highest initial offset was given in test run 8 (Figure A4A), which exhibited the highest initial $c_{B,Lys}$ and therefore more fouling. Test run 7 (Figure 7E) was performed at CF 350 mL/min, which was outside the training space. Both hybrid models predicted the flux of test run 7 (Figure 7E) well, indicating that the models were not necessarily limited by the training space and showed good extrapolation capabilities of the input parameter CF. Test runs 4, 5 and 6 (Figures 6C,E and 7C) exhibited TMPs and

CFs within the training space parameters and all predicted well. The good flux predictions of these test runs showed the excellent interpolation capabilities of the ANN-aided hybrid models.

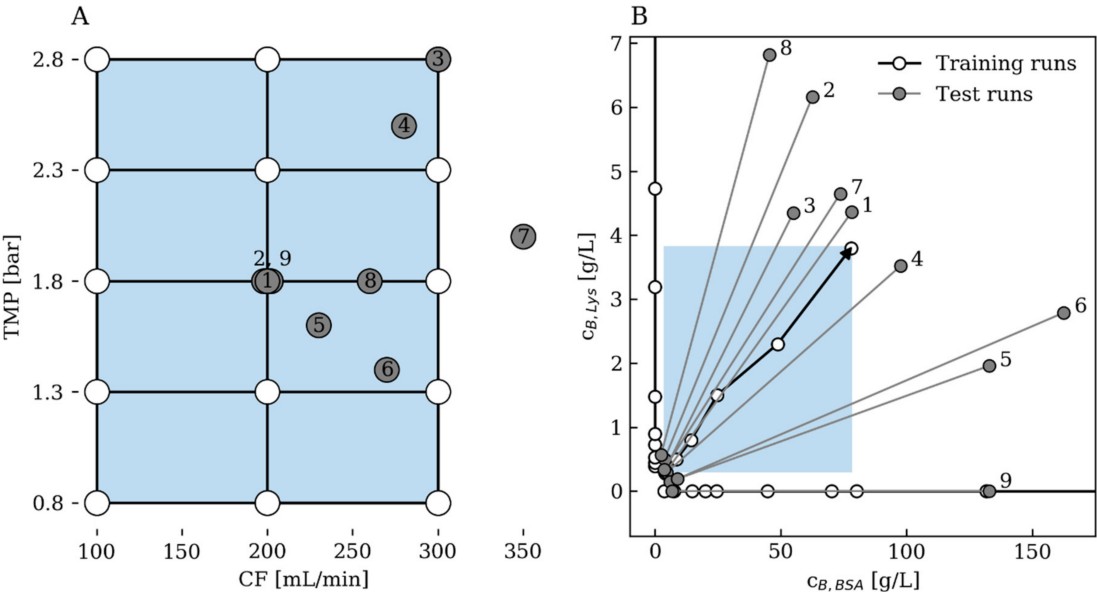

**Figure 5.** Schematic depiction of the training space (blue area) for: (**A**) TMP and CF of training runs (white dots) and test runs (grey dots); (**B**) initial to final $c_{B,BSA}$ and $c_{B,Lys}$ of the test runs (grey dots with grey solid lines) and the covered concentration range of the three training runs (white dots with black solid lines).

The SFM predicted the initial flux and flux evolution inside the training space well (test runs 1, 3 and 4; Figures 6A,C, and A4C). However, for the test runs outside the training space, higher errors were exhibited (test runs 5, 6 and 8; Figures 6E, 7C and A4A). Outside the training space, k and $c_G$ were extrapolated from the training data, which potentially increased flux prediction uncertainty. Furthermore, high lysozyme concentrations also led to higher errors due to stronger fouling over time and not being able to incorporate the second component in the SFM. Here, the SFM underestimated the initial flux drastically (test runs 2 and 8; Figures 7A and A4A). For test run 9 (Figure A4E)—only BSA, no lysozyme—the SFM with k and $c_G$ were exceptionally based on BSA training data (Figure 3C) to allow fair comparison. In this case, the SFM yielded good initial flux predictions, but deviations at the end of the process, while HM 1 and 2 both showed excellent flux prediction over the entire process. On average, the flux prediction error of SFM was 6% NRMSE, while the error of the two hybrid models was 4.1% and 3.9% NRMSE (Figure 8A).

3.3.2. Rejection Factor Prediction for Lysozyme

The rejection factor for lysozyme $R_{Lys}$ increased throughout the UF run, from around 0.6 to almost 1.0, as shown in Figures 6, 7 and A4. The pores became increasingly blocked throughout the UF process, most probably because lysozyme was absorbed in their inner wall, increasing the rejection factor. Results showed that there was no consistent correlation between the TMP and $R_{Lys}$, or CF and $R_{Lys}$ (Figure A5). Therefore, the influence of TMP and CF on $c_{P,Lys}$ was neglected when creating the calibration between UV absorbance and lysozyme permeate concentration. The rejection factor of BSA was 1 for all experiments. The model errors are given in Figure 8B.

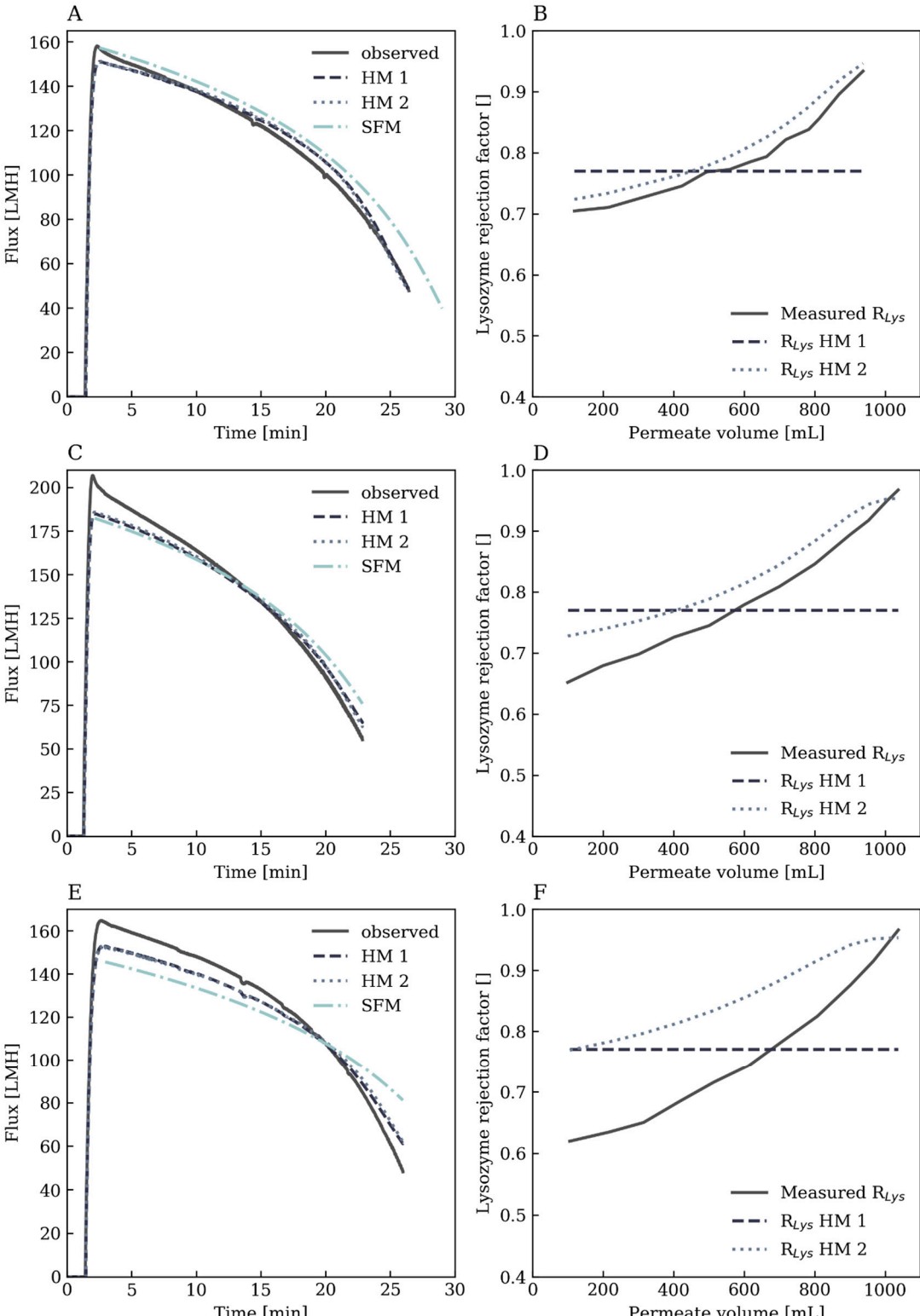

**Figure 6.** Comparison of observed and predicted flux and R_Lys. (**A**) The flux over time of test run 1, (**B**) R_Lys over permeate volume of test run 1, (**C**) the flux over time of test run 4, (**D**) R_Lys over permeate volume of test run 4, (**E**) the flux over time of test run 5, (**F**) R_Lys over permeate volume of test run 5.

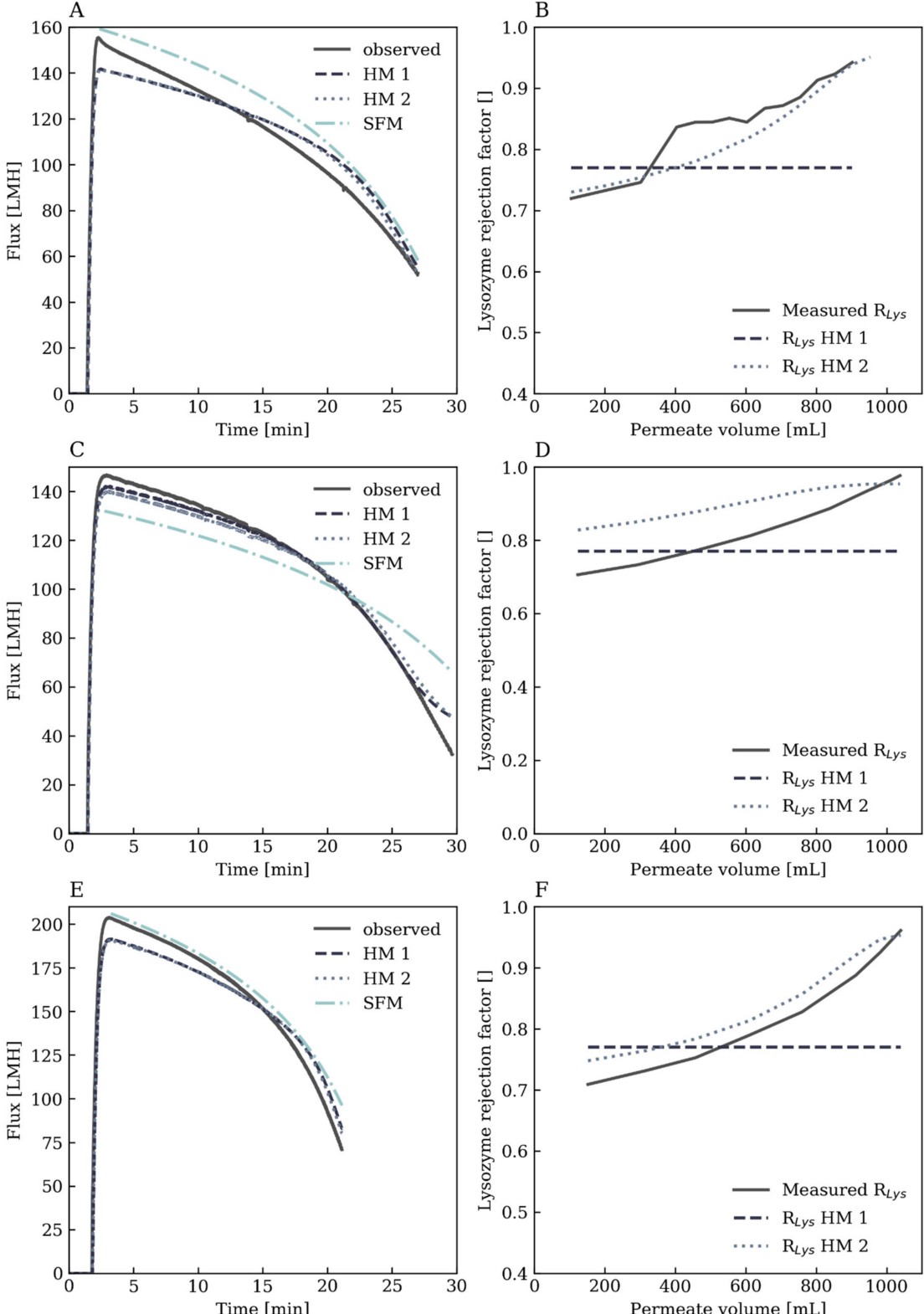

**Figure 7.** Comparison of observed and predicted flux and $R_{Lys}$. (**A**) The flux over time of test run 2, (**B**) $R_{Lys}$ over permeate volume of test run 2, (**C**) the flux over time of test run 6, (**D**) $R_{Lys}$ over permeate volume of test run 6, (**E**) the flux over time of test run 7, (**F**) $R_{Lys}$ over permeate volume of test run 7.

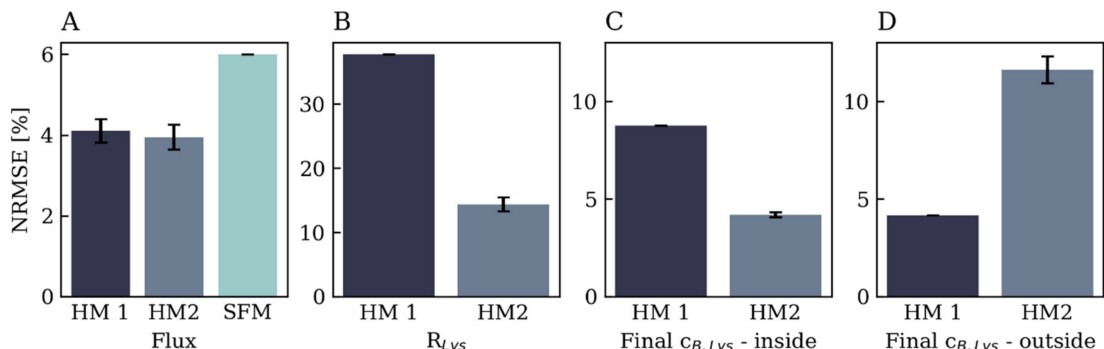

**Figure 8.** Summary of the prediction errors of HM 1, HM 2 and SFM in terms of (**A**) flux, (**B**) $R_{Lys}$, (**C**) final $c_{B,Lys}$ for test runs 1 to 4 with all parameters—TMP, CF, initial $c_{B,BSA}$ and $c_{B,Lys}$—inside the training space and (**D**) final $c_{B,Lys}$ for test runs 5 to 8 performed partly outside the training space.

Hybrid Model 1: Constant Lysozyme Rejection Factor

In HM 1 the rejection factor for lysozyme $R_{Lys}$ was assumed to be constant over time for all test runs, where lysozyme was present and therefore exhibited the largest $R_{Lys}$ error (38% NRMSE) compared to HM 2 (see Figure 8A). All test runs (Figures 6, 7 and A4) show that HM 1 overestimated $R_{Lys}$ at the beginning of all UF runs and underestimated it at the end. The average $R_{Lys}$ based on training data fitted all independently generated test data very well but lacked the ability to adjust to the increasing $R_{Lys}$.

Hybrid Model 2: Dynamic Lysozyme Rejection Factor

In contrast to keeping the rejection factor constant, as in HM 1, a second black box was introduced in HM 2 to predict $R_{Lys}$ dynamically. This prediction was independent of the flux prediction but was based on the same four input parameters, namely TMP, CF, initial $c_{B,BSA}$ and initial $c_{B,Lys}$. The NRMSE of the newly introduced black box was evaluated by comparing the observed $R_{Lys}$ values to the predicted $R_{Lys}$. Since the correlation of $R_{Lys}$ and $V_P$ is quite simple, an ANN with one hidden node was used for $R_{Lys}$ prediction (Figure A1C). For comparison, a multiple linear regression (MLR) model was also tested as an alternative black box, resulting in a less complex hybrid model that required less computation time and facilitated easier interpretability. However, the ANN with one node was chosen instead of the MLR, because of the lower prediction error regarding $R_{Lys}$ and final $c_{B,Lys}$ (see Table A4).

HM 2, with an average $R_{Lys}$ NRMSE of 14%, performed better than HM 1. The improvement was achieved as HM 2 considered the increasing $R_{Lys}$ over the process, which subsequently strongly influenced the final $c_{B,Lys}$ prediction (Section 3.3.3). In test runs 5 and 6 (Figures 6F and 7D) the prediction from HM 2 overestimated $R_{Lys}$. These test runs exhibited a low TMP and high $c_{B,BSA}$. The hybrid model assumed that the CP layer of BSA and fouling due to lysozyme were at an equilibrium, at which the lysozyme transmission was lower than in the test runs, where the CP layer was still building up. Low TMP additionally prolonged the time to reach flux steady state. $R_{Lys}$ of the other test runs 1, 2, 4, 7 and 8 (Figures 6B,D, 7B,F and A4B) were predicted accurately with HM 2.

Even though HM 2 performed better than HM 1 in $R_{Lys}$ prediction, the flux predictions were almost identical (NRMSE 3.9 and 4.1 %). This indicated that they were not affected by small variations or changes in $c_{B,Lys}$.

3.3.3. Endpoint Bulk Concentration

Since $R_{BSA}$ was 1, all models—HM and SFM—predicted the same $c_{B,BSA}$ at the final retentate volume, with an average error of 4.2% (Table A3). BSA did not show membrane fouling and was quantitatively recovered at the end of the process. The predictions of the final $c_{B,Lys}$ varied because

of the different $R_{Lys}$ predictions. The discussion for $c_{B,Lys}$ prediction was divided into the test runs performed strictly inside and outside the training space, since the hybrid models performed differently.

Within the training space—test runs 1−4—HM 1 and HM 2 performed in accordance with the $R_{Lys}$ predictions (Figure 8C). HM 1 exhibited the highest error of 9% since $R_{Lys}$ was not adjusted over the processing time. HM 2 recalculated $R_{Lys}$ with every iteration; its $c_{B,Lys}$ predictions were in good accordance with the measured concentrations, with an NRMSE of 4% and superior to HM 1. Similarly to the $R_{Lys}$, the accuracy of the final $c_{B,Lys}$ prediction benefited from two separately trained black boxes.

In cases where at least one input parameter was outside of the training space—test runs 5−8—HM 1 performed best with an average NRMSE of 4% (Figure 8D). In comparison, HM 2 yielded worse final $c_{B,Lys}$ predictions, exhibiting a three-fold increase in NRMSE (12%). Even though $R_{Lys}$ was updated in HM 2, it was overestimated throughout most of the test runs, leading to higher $c_{B,Lys}$ prediction and a cumulated NRMSE that increased with the duration of the process. In contrast, using HM 1 the initial $R_{Lys}$ over-prediction and under-prediction balanced out and yielded acceptable final $c_{B,Lys}$ predictions.

In summary, the more complex HM 2 showed superior performance within the trained space, which is the case for most modeling applications. It can predict the flux, $R_{Lys}$ and therefore the concentration, of both components at any time point of the process. For predictions outside the trained space, the simpler and more robust HM 1 performed better, giving accurate predictions on flux and the fully retained main component BSA. It can offer valuable insights when exploring parameter ranges if the desired optimal process conditions are not met in the trained space, before it is expanded and used to retrain new hybrid models.

## 4. Conclusions

UF modeling increases process understanding which is key for predicting process performance. The interactions of various components means that mechanistic modeling approaches for multi-component solutions might become very complex and require many experiments.

We developed and compared hybrid models to predict flux, rejection behavior and concentrations for UF of two-component solutions. The models were trained on training experiments that were generated in less than eight hours and tested on independently performed UF runs with varying product and impurity concentrations, TMPs and CFs. We showed that the hybrid model HM 2, with a dynamic impurity rejection factor containing two black boxes, exhibited the best predictions for impurity rejection behavior and final concentration within the trained parameter space and had excellent interpolation properties. The simpler HM 1 yielded stable predictions beyond the trained space, rendering it a valuable tool for extrapolation. Both hybrid models performed similarly well in predicting flux and mimicked product concentration. The SFM with mechanistic parameters exhibited higher flux prediction errors than both hybrid models and could not predict the lysozyme rejection factor and final concentration, since it can only assume a one-component system. Our results show that it is crucial to quantify and incorporate all components, including the impurities, to gain accurate and reliable process models. These variations can be included more easily in the hybrid model approach than in mechanistic models such as SFM, with low experimental effort and no mechanistic parameter adaption required.

A limitation of the presented models is the time-dependent fouling of the mimicked impurity at high initial concentrations. However, at the expected concentration ranges, e.g., after the chromatography capture step, the effect can be neglected.

The proposed hybrid model structure can be used not only for the reliable prediction of final product concentrations, but also of the concentration of various quantifiable classes of impurities. Since impurities are a critical quality attribute (CQA) in many manufacturing bioprocesses, time-resolved concentration predictions help to better understand the process's outcome upfront. Furthermore, by taking adequate measures a potential batch rejection due to high impurity concentration can be avoided. The product and impurities can be measured with online sensors or correlated with

offline analytics using soft sensors. In combination, with closed-loop process controllers, these hybrid models are a valuable tool for increased process understanding and advanced process control.

**Author Contributions:** Conceptualization, M.D.; methodology, M.K.; software, M.K. and I.B.-M.; validation, M.K. and I.B.M.; formal analysis, M.K.; investigation, M.K.; resources, A.D.; data curation, M.K. and I.B.-M.; writing—original draft preparation, M.K.; writing—review and editing, M.K., M.D., A.D.; visualization, M.K.; supervision, A.D. and M.D.; project administration, M.D.; funding acquisition, M.D. All authors have read and agreed to the published version of the manuscript.

**Funding:** This research was funded by the Austrian Research Promotion Agency (FFG), grant number 859219.

**Conflicts of Interest:** The authors declare no conflict of interest.

## Symbols and Abbreviations

| | |
|---|---|
| ANN | artificial neural network |
| BSA | bovine serum albumin |
| CF | cross-flow velocity |
| CP | concentration polarization |
| HM | hybrid model |
| MWCO | molecular weight cutoff |
| NRMSE | normalized root-mean-squared error |
| SEC | size exclusion chromatography |
| SFM | stagnant film model |
| TMP | transmembrane pressure |
| UF | ultrafiltration |
| | |
| A | membrane area [m$^2$] |
| $c_B$ | bulk concentration [g/L] |
| $c_G$ | gel layer concentration [g/L] |
| $c_P$ | permeate concentration [g/L] |
| $c_R$ | retentate concentration [g/L] |
| dt | time increment [s] |
| J | permeate flux [LMH] or [m/s] |
| k | mass transfer coefficient [LMH] |
| $R_{Lys}$ | lysozyme retention coefficient [-] |
| $V_B$ | bulk/reservoir volume [mL] |
| $V_P$ | permeate volume [mL] |

## Appendix A

*Appendix A.1. Neural Network Model Optimization*

To choose the best-suited ANN structure, varying numbers of hidden nodes were tested. Each ANN was trained on the combined training set and validated. Figure A1A gives an overview of the optimal ANN structure including the inputs TMP, CF, $c_{B,BSA}$, $c_{B,Lys}$, and the output permeate flux (in HM 2 a second ANN with $R_{Lys}$ as output was added) with four hidden neurons. The input and output parameters were scaled between 0 and 1 before optimizing the ANN. This step is necessary to have the parameters on the same scale rendering them comparable. Each node in the hidden and output layer in Figure A1A forms a linear equation. As an example, the first hidden node $x_{21}$ is the sum of each multiplication of an input (TMP, CF, $c_{B,BSA}$, $c_{B,Lys}$) and the corresponding weight ($w^1_{11}$, $w^1_{21}$, $w^1_{31}$, $w^1_{41}$) multiplied with the bias ($b_1$) of the entire hidden layer.

$$x_{21} = b_1\left(w^1_{11}TMP_{scaled} + w^1_{21}CF_{scaled} + w^1_{31}c_{B,BSA,scaled} + w^1_{41}c_{B,\,Lys,scaled}\right)$$

To determine the values for the weights and biases that result in the desired prediction the model is optimized in multiple epochs. As a first step, the weights and biases are randomly chosen and the first prediction with inputs from a given training set is performed. Since the weights and biases are not optimized the flux prediction will be of poor quality and the prediction error will be high. Using the desired output from the training set, the ANN is calculated backward which results in inputs parameters that fit the prediction. The error between the real and the backward calculated inputs is estimated and used to update the according to weights and biases.

This optimization can be performed with different algorithms—in this publication we chose MATLAB's trainbr function which employs Bayesian regularization, which is an adaptation of the Levenberg-Marquardt optimizer and minimizes the squared errors and weights. Once the weights and biases are optimized, the ANN structure is defined and applied to the test sets and will predict the same results for a given set of inputs.

The presented ANN structure (Figure A1A) was determined after screening a wide number of nodes in the hidden layer from one to seven, with the corresponding NRMSE being recorded and averaged. We chose an ANN with four hidden nodes for all hybrid model structures (Figure A1B) because it exhibited the lowest average error and standard deviation. Structures with less than four nodes resulted in under-fitted models. More hidden nodes led to an over-fitting of the training data and higher prediction error and standard deviation. In HM 2 the $R_{Lys}$ black box ANN was optimized in the same way with 1 hidden node yielding the lowest error. The tested ANNs consisted of one hidden layer with a sigmoid activation function and linear activation functions in both the input and output layer. The inputs were normalized between 0 and 1.

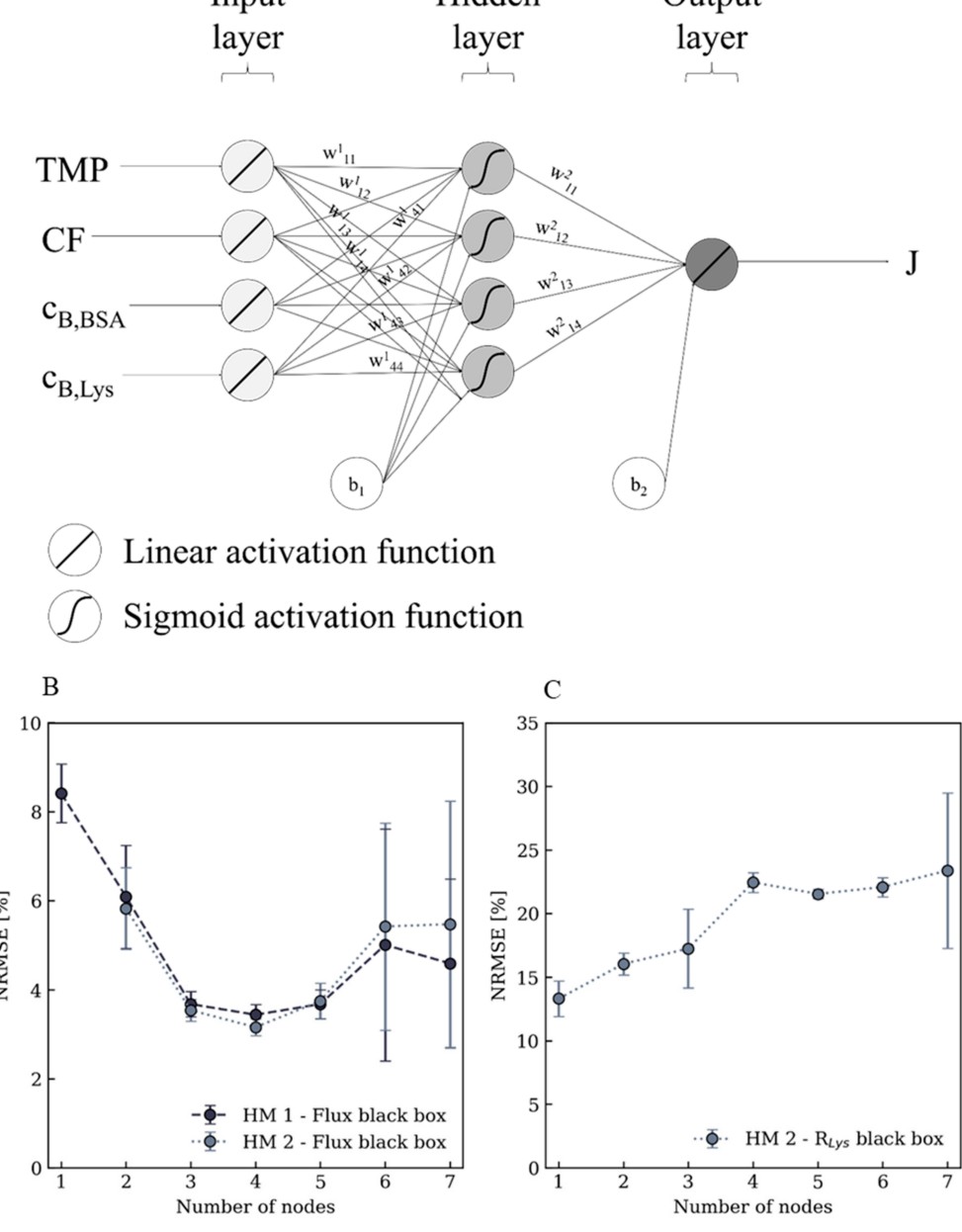

**Figure A1.** (**A**) Structure of the ANN including input, output and hidden layer, number of hidden nodes and activation functions. (**B**,**C**) Optimization of the ANN structure. (**B**) The NRMSE is plotted over the number of neurons in the hidden layer of the ANN of the permeate flux predicting black box. (**C**) The NRMSE of the $R_{Lys}$ black box HM 2.

*Appendix A.2. Experimental Data Summary*

Table A1 gives a summary of the measured $c_{B,BSA}$, and $c_{B,Lys}$ for the three training sets containing BSA with lysozyme, only BSA and only lysozyme. Training set 2 and 3 consisted of two parts with different starting $c_{B}$s to cover a wide $c_{B}$ space. For each $c_{B}$ the TMP and CF were increased stepwise in three minutes intervals. Afterward, $c_{B}$ was increased by concentrating the sample and removing the generated permeate from the reservoir.

**Table A1.** Summary of all training data including measured $c_{B,BSA}$, $c_{B,Lys}$, and observed flux.

| Training Set | Observation | $c_{B,Lys}$ [g/L] | $c_{B,BSA}$ [g/L] | Flux at TMP 0.8 bar [LMH] | Flux at TMP 1.3 bar [LMH] | Flux at TMP 1.8 bar [LMH] | Flux at TMP 2.3 bar [LMH] | Flux at TMP 2.8 bar [LMH] |
|---|---|---|---|---|---|---|---|---|
| | 100 | 0.28 | 3.77 | 87.6 | 113.3 | 121.0 | 121.5 | 119.1 |
| | | 0.5 | 8.48 | 78.0 | 93.3 | 96.3 | 95.6 | 93.4 |
| | | 0.76 | 14.38 | 70.1 | 80.5 | 81.6 | 80.0 | 77.7 |
| | | 1.51 | 24.68 | 60.7 | 67.5 | 67.5 | 65.9 | 63.9 |
| | | 2.3 | 48.72 | 46.8 | 50.7 | 50.1 | 48.6 | 47.1 |
| | | 3.81 | 77.93 | 34.0 | 36.6 | 36.2 | 35.1 | 34.0 |
| 1 | 200 | 0.28 | 3.77 | 92.9 | 136.3 | 157.6 | 164.5 | 164.2 |
| | | 0.5 | 8.48 | 89.3 | 121.2 | 131.9 | 132.7 | 130.3 |
| | | 0.76 | 14.38 | 82.8 | 106.5 | 112.3 | 111 | 108.3 |
| | | 1.51 | 24.68 | 74.2 | 91.1 | 93.5 | 91.6 | 88.8 |
| | | 2.3 | 48.72 | 59.2 | 68.6 | 69.2 | 67.1 | 64.7 |
| | | 3.81 | 77.93 | 43.3 | 49.3 | 49.4 | 47.8 | 46.2 |
| | 300 | 0.28 | 3.77 | 92.0 | 143.4 | 178.3 | 194.8 | 199.8 |
| | | 0.5 | 8.48 | 90.2 | 133.8 | 155.6 | 161.5 | 161.1 |
| | | 0.76 | 14.38 | 85.6 | 121.5 | 135.2 | 136.9 | 134.7 |
| | | 1.51 | 24.68 | 78.8 | 106.2 | 114.0 | 113.4 | 110.7 |
| | | 2.3 | 48.72 | 65.1 | 82.2 | 85.1 | 83.2 | 80.6 |
| | | 3.81 | 77.93 | 48.4 | 59.2 | 60.4 | 58.9 | 56.9 |
| | 100 | 3.19 | - | 97.6 | 150.1 | 195.1 | 232.6 | 260.6 |
| | | 4.73 | - | 97.9 | 146.5 | 186.1 | 218.2 | 244.8 |
| | | 8.11 | - | 97.6 | 144.7 | 181.1 | 209.5 | 235.8 |
| | | 11.99 | - | 96.8 | 142.2 | 176.8 | 205.2 | 231.8 |
| | | 23.79 | - | 95.0 | 137.5 | 171.0 | 199.1 | 221.8 |
| | | 32.95 | - | 85.7 | 124.9 | 152.3 | 172.4 | 187.2 |
| 2a | 200 | 0.39 | - | 98.3 | 151.9 | 196.6 | 233.6 | 264.2 |
| | | 0.44 | - | 99 | 150.8 | 192.2 | 224.3 | 252.4 |
| | | 0.53 | - | 98.3 | 148.7 | 187.9 | 218.9 | 244.8 |
| | | 0.72 | - | 98.3 | 146.9 | 184.7 | 215.3 | 241.6 |
| | | 0.9 | - | 96.8 | 143.3 | 178.9 | 208.4 | 232.9 |
| | | 1.48 | - | 89.3 | 128.9 | 160.9 | 186.5 | 204.1 |
| | 300 | 3.19 | - | 97.2 | 152.28 | 198.36 | 236.52 | 267.84 |
| | | 4.73 | - | 97.56 | 150.84 | 195.48 | 229.68 | 258.12 |
| | | 8.11 | - | 96.84 | 149.4 | 192.24 | 225 | 252.36 |
| | | 11.99 | - | 96.84 | 148.32 | 189.36 | 222.12 | 249.84 |
| | | 23.79 | - | 96.12 | 145.08 | 183.6 | 215.28 | 240.84 |
| | | 32.95 | - | 86.76 | 129.96 | 165.6 | 192.6 | 212.76 |
| | 100 | 3.19 | - | 86.0 | 113.8 | 121.0 | 115.6 | 105.5 |
| | | 4.73 | - | 67.3 | 81.0 | 84.2 | 82.4 | 79.9 |
| | | 8.11 | - | 59.8 | 69.8 | 71.6 | 70.6 | 68.8 |
| | | 11.99 | - | 55.1 | 62.6 | 63.4 | 61.9 | 60.5 |
| | | 23.79 | - | 45.7 | 50.0 | 49.7 | 47.9 | 46.8 |
| | | 32.95 | - | 36.0 | 37.8 | 36.4 | 35.3 | 34.2 |
| 2b | 200 | 3.19 | - | 88.2 | 121.7 | 135.7 | 137.9 | 134.3 |
| | | 4.73 | - | 74.5 | 96.1 | 105.8 | 108.4 | 108.0 |
| | | 8.11 | - | 67.7 | 85.7 | 92.9 | 94.3 | 93.6 |
| | | 11.99 | - | 63.0 | 78.5 | 83.9 | 84.2 | 82.8 |
| | | 23.79 | - | 54.0 | 64.8 | 67.0 | 65.5 | 63.4 |
| | | 32.95 | - | 43.9 | 50.0 | 49.7 | 47.9 | 46.8 |
| | 300 | 3.19 | - | 85.0 | 121.0 | 140.0 | 148.7 | 151.2 |
| | | 4.73 | - | 74.9 | 102.6 | 117.7 | 125.6 | 128.5 |
| | | 8.11 | - | 69.5 | 93.2 | 105.8 | 111.6 | 113.0 |
| | | 11.99 | - | 65.5 | 87.5 | 97.6 | 100.8 | 100.8 |
| | | 23.79 | - | 57.2 | 73.8 | 79.2 | 79.6 | 78.1 |
| | | 32.95 | - | 50.0 | 62.6 | 64.8 | - | - |

**Table A1.** *Cont.*

| Training Set | Observation | $c_{B,Lys}$ [g/L] | $c_{B,BSA}$ [g/L] | Flux at TMP 0.8 bar [LMH] | Flux at TMP 1.3 bar [LMH] | Flux at TMP 1.8 bar [LMH] | Flux at TMP 2.3 bar [LMH] | Flux at TMP 2.8 bar [LMH] |
|---|---|---|---|---|---|---|---|---|
| 3a | 100 | - | 3.65 | 90.8 | 119.6 | 131.9 | 136.5 | 138.3 |
| | | - | 7.45 | 84.9 | 106.8 | 114.8 | 117.4 | 117.2 |
| | | - | 14.65 | 76.6 | 92.4 | 97.0 | 97.3 | 96.4 |
| | | - | 20.06 | 68.2 | 82.1 | 85.5 | 85.1 | 83.3 |
| | | - | 24.4 | 67.2 | 78.7 | 82.0 | 82.0 | 80.8 |
| | | - | 44.78 | 53.2 | 61.4 | 63.2 | 62.9 | 61.6 |
| | 200 | - | 3.65 | 97.9 | 144 | 171.7 | 185.4 | 191.5 |
| | | - | 7.45 | 96.1 | 135 | 154.4 | 162 | 164.5 |
| | | - | 14.65 | 90.4 | 119.9 | 132.1 | 135.7 | 135 |
| | | - | 20.06 | 81.4 | 108.4 | 118.4 | 119.9 | 118.1 |
| | | - | 24.4 | 82.1 | 104.8 | 113 | 114.5 | 113 |
| | | - | 44.78 | 67.3 | 82.4 | 86.8 | 86.8 | 85 |
| | 300 | - | 3.65 | 98.4 | 151.4 | 190.8 | 214.9 | 228.5 |
| | | - | 7.45 | 98.6 | 147.2 | 178.4 | 194.4 | 201.0 |
| | | - | 14.65 | 93.8 | 134.4 | 155.9 | 164.9 | 167.2 |
| | | - | 20.06 | 87.5 | 124.9 | 143.4 | 149.2 | 149.0 |
| | | - | 24.4 | 87.6 | 120.8 | 136.0 | 141.1 | 141.3 |
| | | - | 44.78 | 74.6 | 97.9 | 106.1 | 107.9 | 106.8 |
| 3b | 100 | - | 70.12 | 40.7 | 46.4 | 47.3 | 46.6 | 45.5 |
| | | - | 80.15 | 39.4 | 45.5 | 46.9 | 46.6 | 45.8 |
| | | - | 131.69 | 24.0 | 28.8 | 29.9 | 29.6 | 29.0 |
| | | - | 179.95 | 13.6 | 18.1 | 19.6 | 19.7 | 19.7 |
| | | - | 231.67 | 6.3 | 10.9 | 13.1 | 13.8 | 13.7 |
| | | - | 277.25 | 0.0 | 5.5 | 7.7 | 8.8 | 9.4 |
| | 200 | - | 70.12 | 53.6 | 64.1 | 66.2 | 65.5 | 63.4 |
| | | - | 80.15 | 50.8 | 61.2 | 63.7 | 63.4 | 62.6 |
| | | - | 131.69 | 31.8 | 38.9 | 40.3 | 40 | 39.2 |
| | | - | 179.95 | 17.3 | 23.5 | 25.7 | 26 | 25.7 |
| | | - | 231.67 | 7.2 | 13.8 | 16.3 | 17.4 | 17.5 |
| | | - | 277.25 | 5.9 | 9.4 | 10.8 | - | - |
| | 300 | - | 70.12 | 62.6 | 79.0 | 83.3 | 82.6 | 63.5 |
| | | - | 80.15 | 58.2 | 73.3 | 78.0 | 78.3 | 62.5 |
| | | - | 131.69 | 37.1 | 46.8 | 49.5 | 49.3 | 39.2 |
| | | - | 179.95 | 19.5 | 27.5 | 30.3 | 30.9 | 25.7 |
| | | - | 231.67 | - | - | - | - | 17.5 |
| | | - | 277.25 | - | - | - | - | - |

Table A2 summarized the parameters chosen for the test experiments: TMP, CF, initial $c_B$, and measured final $c_B$ of BSA and lysozyme.

**Table A2.** Summary of the test sets with varying initial $c_B$s, CF, and TMP.

| Test Set Number | TMP [bar] | CF [mL/min] | Initial $c_{B,BSA}$ [g/L] | Final $c_{B,BSA}$ [g/L] | Initial $c_{B,Lys}$ [g/L] | Final $c_{B,Lys}$ [g/L] |
|---|---|---|---|---|---|---|
| 1 | 1.8 | 200 | 4.00 | 78.11 | 0.28 | 4.36 |
| 2 | 1.8 | 200 | 3.79 | 62.48 | 0.50 | 6.16 |
| 3 | 2.8 | 300 | 3.82 | 54.95 | 0.32 | 4.35 |
| 4 | 2.5 | 280 | 4.56 | 97.59 | 0.28 | 3.52 |
| 5 | 1.6 | 230 | 5.97 | 132.81 | 0.15 | 1.96 |
| 6 | 1.4 | 270 | 8.80 | 162.45 | 0.19 | 2.79 |
| 7 | 2.0 | 350 | 3.62 | 73.62 | 0.34 | 4.65 |
| 8 | 1.8 | 260 | 2.38 | 45.42 | 0.57 | 6.82 |
| 9 | 1.8 | 200 | 6.68 | 132.70 | 0.00 | 0.00 |

Figure A2 gives the entire training data sets at CF 200 mL/min.

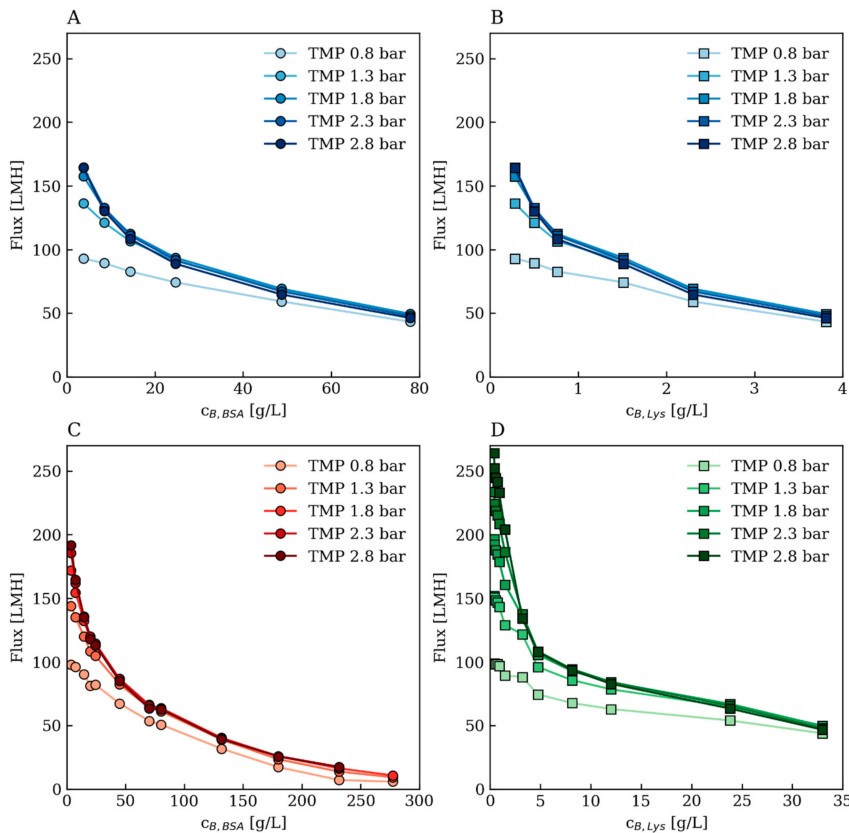

**Figure A2.** Training data sets including different protein concentrations, TMP at CF 200 mL/min. (**A**,**B**) Multi-component training set containing BSA and lysozyme (blue) in the same solution and one component solution of (**C**) BSA (red) and (**D**) lysozyme (green).

Figure A3 gives the correlation curve to calculate $R_{Lys}$ from the UV absorbance at 280 nm using Equation (2). The curve was calculated using the lysozyme training data (Table A1, Training set 2) and exhibits a correlation coefficient of 0.97.

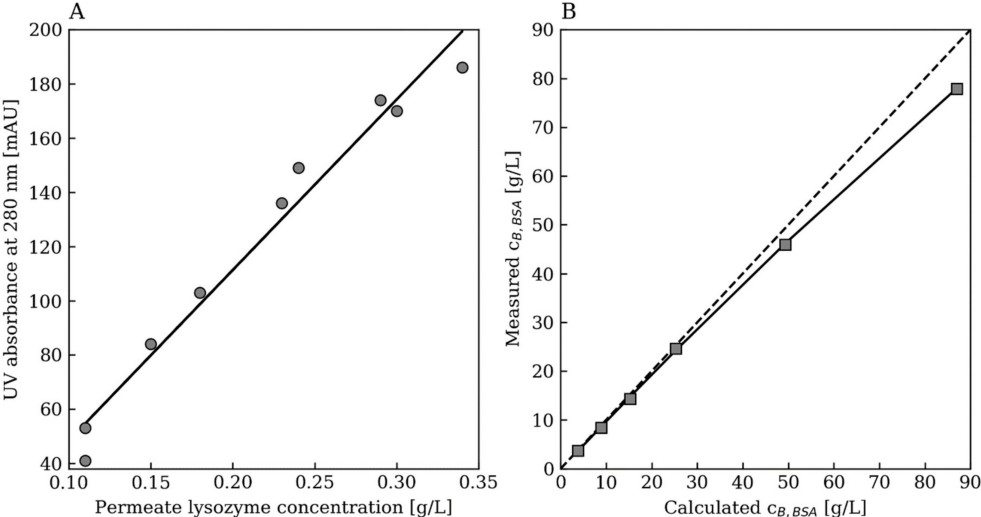

**Figure A3.** (**A**) Calibration of UV absorbance at 280 nm on the permeate side of the membrane versus $c_{P,Lys}$ of the lysozyme training set measured with HP-SEC ($R^2$ = 0.97). (**B**) Deviations between calculated and measured $c_{B,BSA}$ with an $R^2$ of 0.9997, including the identity line (dotted line). The concentration steps were performed at TMP 1.8 bar and CF 200 mL/min.

The applied polynomial function to correct for the CP layer was:

$$c_{meas,B,BSA} = -0.0012 \cdot c_{calc,B,BSA}{}^2 + 1.0063 \cdot c_{calc,B,BSA} \tag{A1}$$

This function is specific to the protein and the membrane and must be adapted for new protein-membrane combinations.

### Appendix A.3. Further Modeling Results

Figure A4 gives the flux predictions of all three model structures for test runs 8, 3, and 9.

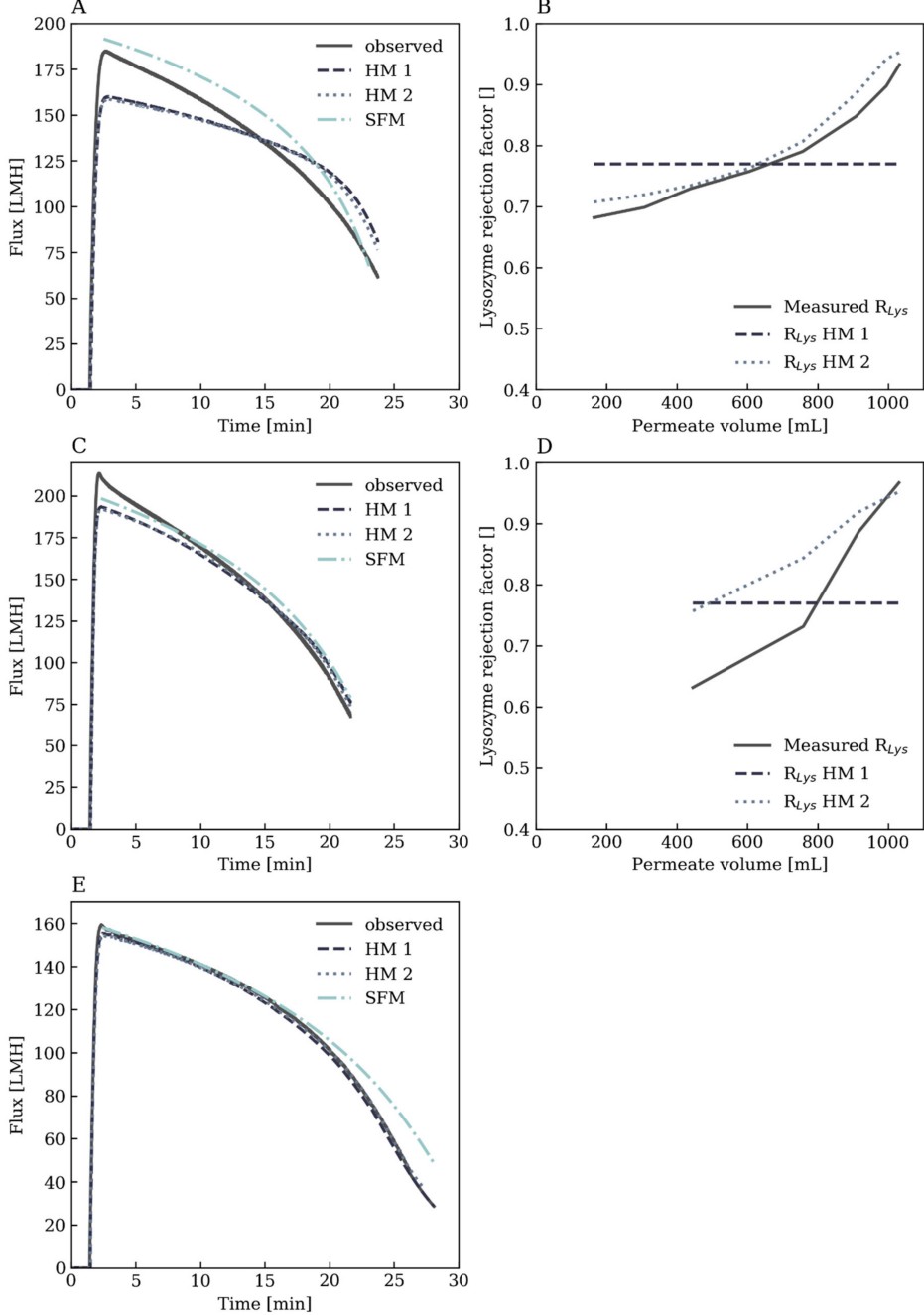

**Figure A4.** Comparison between observed and predicted flux and $R_{Lys}$. (**A**) The flux of test run 8 over time, (**B**) $R_{Lys}$ over permeate volume of test run 8, (**C**) the flux over time of test run 3, (**D**) $R_{Lys}$ over permeate volume of test run 3, (**E**) the flux over time of test run 9.

Table A3 contains the prediction errors of HM 1 and 2 regarding flux, $R_{Lys}$, and final $c_{B,BSA}$, and $c_{B,Lys}$ prediction. The $c_{B,BSA}$ predictions are identical for both hybrid models because $R_{BSA}$ is 1. Test run 9 contained no lysozyme, therefore no $R_{Lys}$ and $c_{B,Lys}$ error was calculated. The HM 2 NRMSE for final $c_{B,Lys}$ show clear differences between test run 1−4 (input parameters inside the trained space) and test run 5−8 (at least one input parameter outside the trained space).

**Table A3.** Summary of the test sets with varying initial $c_B$, CF, and TMP, and the number of samples taken to calculate the observed R.

| Test Set Number | NRMSE Flux [%] | | | NRMSE $R_{Lys}$ [%] | | NRMSE final $c_{B,Lys}$ [%] | | NRMSE Final $c_{B,BSA}$ [%] | | |
|---|---|---|---|---|---|---|---|---|---|---|
| | HM 1 | HM 2 | SFM | HM 1 | HM 2 | HM 1 | HM 2 | HM 1 | HM 2 | SFM |
| 1 | 2.3 ± 0.3 | 2.1 ± 0.3 | 5.3 | 32.5 ± 0.0 | 6.2 ± 0.2 | 7.4 ± 0.0 | 4.1 ± 0.0 | 4.6 | 4.6 | 4.6 |
| 2 | 4.3 ± 0.2 | 4.1 ± 0.1 | 7.2 | 39.7 ± 0.0 | 14.7 ± 2.2 | 7.9 ± 0.0 | 4.9 ± 0.3 | 3.8 | 3.8 | 3.8 |
| 3 | 3.6 ± 0.2 | 3.2 ± 0.2 | 3.0 | 40.3 ± 0.0 | 20.4 ± 0.5 | 13.0 ± 0.0 | 4.3 ± 0.0 | 2.2 | 2.2 | 2.2 |
| 4 | 3.9 ± 0.3 | 3.7 ± 0.2 | 5.0 | 34.9 ± 0.0 | 6.9 ± 0.6 | 6.7 ± 0.0 | 3.4 ± 0.2 | 0.7 | 0.7 | 0.7 |
| 5 | 4.8 ± 0.3 | 4.9 ± 0.4 | 8.5 | 32.6 ± 0.0 | 25.7 ± 1.3 | 3.2 ± 0.0 | 11.9 ± 0.5 | 8.7 | 8.7 | 8.7 |
| 6 | 3.4 ± 0.3 | 3.3 ± 0.5 | 9.3 | 45.3 ± 0.0 | 24.5 ±1.3 | 7.6 ± 0.0 | 12.1 ± 0.4 | 0.1 | 0.1 | 0.1 |
| 7 | 4.7 ± 0.3 | 4.5 ± 0.4 | 4.4 | 40.6 ± 0.0 | 6.2 ±0.7 | 3.9 ± 0.0 | 9.4 ± 0.3 | 6.6 | 6.6 | 6.6 |
| 8 | 8.2 ± 0.2 | 8.0 ± 0.3 | 6.3 | 35.6 ± 0.0 | 10.0 ±2.0 | 1.9 ± 0.0 | 13.0 ± 1.6 | 7.2 | 7.2 | 7.2 |
| 9 | 1.7 ± 0.5 | 1.7 ± 0.4 | 4.9 | 0.0 | 0.0 | 0.0 | 0.0 | 4.6 | 4.6 | 4.6 |

Figure A5 showed that there was no consistent correlation between the TMP and $R_{Lys}$, and CF and $R_{Lys}$.

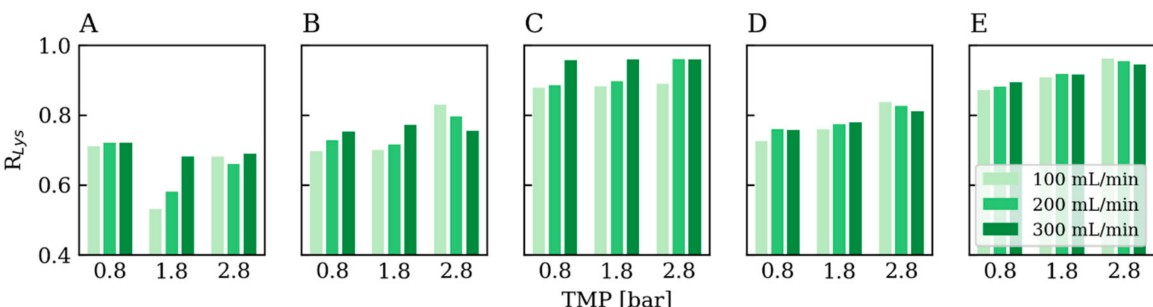

**Figure A5.** $R_{Lys}$ and CF 100, 200, and 300 mL/min versus TMP for increasing bulk concentrations (**A**) 3.19 g/L, (**B**) 4.73 g/L, (**C**) 8.11 g/L, (**D**) 11.99 g/L, and (**E**) 23.79 g/L of the lysozyme training set.

In Table A4 two HM 2 models employing different black box types for $R_{Lys}$ prediction were compared. An ANN with one hidden node in one hidden layer performed better than MLR with linear and interaction terms when assessed for to final $c_{B,Lys}$ and $R_{Lys}$ predictions. Regarding flux prediction, both models perform equally.

**Table A4.** HM 2 black box for $R_{Lys}$ prediction. Comparison between an ANN with one hidden node and MLR regarding, final $c_{B,Lys}$, $R_{Lys}$ and flux prediction error.

| Test Run Number | NRMSE Final $c_{B,Lys}$ [%] | | NRMSE $R_{Lys}$ [%] | | NRMSE Flux [%] | |
|---|---|---|---|---|---|---|
| | 1-node ANN | MLR | 1-node ANN | MLR | 1-node ANN | MLR |
| 1 | 4.1 | 5.7 | 6.2 | 9.9 | 2.1 | 2.3 |
| 2 | 4.9 | 6.4 | 14.7 | 26.9 | 4.1 | 3.9 |
| 3 | 4.3 | 4.3 | 20.4 | 20.9 | 3.2 | 3.3 |
| 4 | 3.4 | 4.7 | 6.9 | 21.1 | 3.7 | 3.8 |
| 5 | 11.9 | 6.4 | 25.7 | 34.5 | 4.9 | 5.1 |
| 6 | 12.1 | 11.6 | 24.5 | 69.4 | 3.3 | 3.5 |
| 7 | 9.4 | 16.1 | 6.2 | 20.0 | 4.5 | 4.4 |
| 8 | 13.0 | 38.4 | 10.0 | 55.5 | 8.0 | 7.5 |
| Average | 7.9 | 11.7 | 14.3 | 32.3 | 3.9 | 4.0 |

Table A5 summarized the mass transfer coefficient k and gel concentration $c_G$ for flux prediction using the SFM. The SFM can be set up for a one-component solution only and since the BSA concentration was 4 to 46 times higher than the lysozyme concentration, BSA was chosen as the modeled component. k and $c_G$ for BSA were calculated for a BSA one-component solution (Table A5 left) and two-component solution (Table A5 right) containing BSA with lysozyme. k from two-component solution was generally lower than for one-component

due to the fouling properties of lysozyme on the cellulose-based membrane, which reduced the transmembrane mass transfer.

**Table A5.** Mass transfer coefficient k and gel concentration $c_G$ for SFM based on BSA (left) and BSA with lysozyme (right).

| | | k based on BSA | | | | | k based on BSA with lysozyme | | |
|---|---|---|---|---|---|---|---|---|---|
| | | Feedflow [mL/min] | | | | | Feedflow [mL/min] | | |
| | | 100 | 200 | 300 | | | 100 | 200 | 300 |
| | 0.8 | 47.36 | 36.23 | 31.63 | | 0.8 | 17.61 | 33.84 | 14.03 |
| TMP | 1.3 | 54.67 | 41.97 | 32.33 | TMP | 1.3 | 25.03 | 36.12 | 27.92 |
| [bar] | 1.8 | 54.51 | 42.14 | 30.13 | [bar] | 1.8 | 27.59 | 36.75 | 39.06 |
| | 2.3 | 53.40 | 41.63 | 28.99 | | 2.3 | 28.11 | 38.22 | 44.73 |
| | 2.8 | 53.75 | 42.97 | 27.95 | | 2.8 | 27.70 | 38.58 | 46.84 |
| | | $c_G$ based on BSA | | | | | $c_G$ based on BSA with lysozyme | | |
| | | Feedflow [mL/min] | | | | | Feedflow [mL/min] | | |
| | | 100 | 200 | 300 | | | 100 | 200 | 300 |
| | 0.8 | 277.83 | 303.41 | 279.25 | | 0.8 | 665.40 | 280.12 | 4421.42 |
| TMP | 1.3 | 302.79 | 330.45 | 323.30 | TMP | 1.3 | 355.06 | 312.56 | 887.24 |
| [bar] | 1.8 | 322.39 | 345.88 | 355.30 | [bar] | 1.8 | 288.49 | 277.52 | 419.22 |
| | 2.3 | 332.29 | 353.74 | 369.46 | | 2.3 | 264.69 | 273.21 | 304.56 |
| | 2.8 | 323.99 | 327.45 | 378.28 | | 2.8 | 256.72 | 252.80 | 263.97 |

Figure A6 shows an exemplary plot to graphically calculate k (negative slope) and $c_G$ (intercept with the abscissa):

$$J = -k \cdot c_B + k \cdot c_G$$

with −k (negative mass transfer rate) being the slope and $k \cdot c_G$ being the intercept with the ordinate. The latter is divided by k, resulting in the gel layer concentration $c_G$.

At the lowest two TMPs (0.8 and 1.3 bar) the flux-ln($c_{B,BSA}$)-curve is not linear, since the is in the pressure-dependent region. In this case, only the points in the linear range were taken for calculating k and $c_G$. For low TMPs and high CFs the flux is pressure dependent. In these cases, the SFM flux prediction will always overestimate the flux. The test runs, however, were performed at a TMP of 1.4 bar or higher, and therefore in the pressure independent flux region.

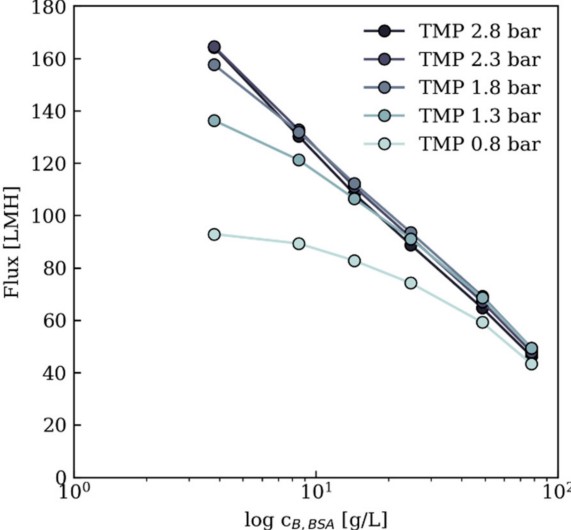

**Figure A6.** Permeate flux vs. logarithmic $c_{B,BSA}$ to estimate the mass transfer coefficient k and gel concentration $c_G$ for the stagnant film theory (SFM). Data recorded at CF 200 mL/min.

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
