# Peer review of "Hybrid Modeling for Simultaneous Prediction of Flux, Rejection Factor and Concentration in Two-Component Crossflow Ultrafiltration"

_processes, doi:10.3390/pr8121625_

Round 1

Reviewer 1 Report

The revised manuscript fully answered the questions that I had before. As a result, the overall quality of the manuscript has been significantly enhanced. Now the manuscript deserves to be published.

Author Response

Thank you very much for your positive feedback and your support to improve our mansucript

Reviewer 2 Report

The work presents a hybrid-modelling strategy for properties prediction in a ultrafiltration process. While the paper is well organized and written. It lacks of a proper description of the models identification, for both empirical and phenomenological models, for example how the ANN hyperparameters were identified? Furthermore, it is difficult to understand the work's contributions when compared with previous works.

Author Response

Question 1: It lacks of a proper description of the models identification, for both empirical and phenomenological models, for example how the ANN hyperparameters were identified?

Answer: In the Appendix we have added a detailed explanation of how the hyperparameters (weights and biases) of the neural network are optimized throughout the training part. It includes the mathematical function behind a hidden node, how the input parameters are processed, how the hyperparameters are optimized in steps (epochs), and what the MATLAB function that we used for optimization uses. Those mentioned steps (yellow highlighted in the Appendix) are all included and performed automatically by the MATLAB function trainbr. They can be customized by the user, but the actual calculation is performed in the background. The actively chosen parameters (inputs, outputs, number of hidden neurons, activation functions) were already mentioned in the Appendix before.

Regarding the stagnant film model (SFM) we added some sentences in the main draft, to explain that the solution to find the SFM parameters k and cG is a geometrical solution (using slope and intercept). In the appendix, we added the derived SFM equation with a detailed explanation of each parameter of the adjusted linear equation.

We hope that the added paragraph clarifies the neural network optimization and gives a bigger picture of how machine learning algorithms like neural networks are optimized automatically. Additionally, we added some clarifying sentences for SFM.

Thank you for your valuable feedback and your support to further improve our manuscript.

Question 2: Furthermore, it is difficult to understand the work's contributions when compared with previous works.

Answer: We added a paragraph at the end of the introduction and conclusion to highlight the additional benefits of this work’s modeling approach. The findings are interesting in two ways: from the modeling and the application perspective. From the modeling perspective, we have shown that hybrid models can cope and predict values from more inputs than before, meaning that more process knowledge (process parameters) can be incorporated in a fast way by simply training the model on them.

From the application perspective, knowing the impurity concentration and evolution beforehand helps to take adequate measures upfront. Since impurities such as host cell proteins are critical quality attributes, they must be depleted to a certain range during the process. This is typically evaluated using quality-by-testing approaches after the process. Upfront knowledge based on our model precictions potentially reduces the risk of batch rejection and adds another layer of security.

Round 2

Reviewer 2 Report

The authors have addressed the previous concern, clarifying the issues. Therefore I recommend the manuscript publications.